# Exploring Effective Terminal State: A Null-Model-Guided Graph Diffusion Model

## Abstract

Graph diffusion models have shown promise in generating complex networks, but they often suffer from two critical limitations: On the one hand, terminating the forward diffusion in pure Gaussian noise graph erases the intrinsic structural signatures of the original network, leading to sub-optimal generative outcomes. On the other hand, the unconstrained diffusion trajectory progressively obliterates topological characteristics, resulting in complete structural degradation. To address these issues, we propose **Null-Model-Guided Graph Diffusion (NMG-GD)**, a principled framework with tailored designs for graph generation. First, we claim that traditional isotropic priors (e.g., Gaussian or fully structured graphs) distort salient topological features. Instead, we adopt a null-model distribution as the forward diffusion endpoint, which explicitly preserves critical network statistics such as degree sequences and clustering coefficients—ensuring global consistency. Second, we derive a null-model-guided continuous-time stochastic differential equation (SDE) and introduce the Position-enhanced Graph Score Network (PGSN). PGSN ingests both continuous and quantized adjacencies, fusing random-walk, shortest-path and null-model cues in a permutation-equivariant encoder,which can significantly elevates sample quality. Extensive experiments on three public datasets (including social and biological networks) demonstrate that NMG-GD achieves state-of-the-art performance. It shows the significant advantages in structural similarity and generation efficiency.

## 1 Introduction

Graph generation has wide applications in various fields such as social networks (Davies & Ajmeri, 2022), recommendation systems (Wu et al., 2022), rug synthesis (Yang et al., 2024b), and protein modeling (Liu et al., 2023). Traditional methods for graph generation date back to random graph models (Watts & Strogatz, 1998; Albert & Barabási, 2002; Erdös & Rényi, 2006), which capture limited graph statistic properties. Recent deep graph generative models leverage neural networks to directly learn graph structure distributions effectively. Prominent paradigms include models based on variational autoencoders (VAE) (Kipf & Welling, 2016; Simonovsky & Komodakis, 2018; Liu et al., 2019b), models based on generative adversarial networks (GAN) (Bojchevski et al., 2018; Cao & Kipf, 2022), models based on recurrent neural networks (RNN) (You et al., 2018), flow-based models (Zang & Wang, 2020; Shi et al., 2020; Luo et al., 2021; Liu et al., 2019a), and autoregressive models (Li et al., 2018; You et al., 2018; Liao et al., 2020; Dai et al., 2020; Chen et al., 2021).

Diffusion models have emerged as a rapidly developing technique in computer vision and recently achieved state-of-the-art performance in enhancing generative capabilities in the image domain (Cao & Kipf, 2022; Yang et al., 2024a). It gradually perturbs the data distribution during the forward diffusion process and then learns to recover the data distribution from noise through the reverse diffusion process. Motivated by the compelling generative performance of diffusion models, a growing body of work has begun to adapt these techniques to graph data and derived graph diffusion models can be categorized into two main types: discrete denoising diffusion models and continuous denoising diffusion models. The core idea of discrete denoising diffusion models is to introduce noise via a sequence of discrete graph edits, while continuous denoising diffusion models add Gaussian noise to the node features and graph adjacency matrices. In the reverse generation phase, a graph neural network is developed to predict the original clean graph from the noisy input.

Despite promising results on some datasets, current graph diffusion models usually set the terminal state to either an isotropic Gaussian graph or a fully structured one(complete graphs or empty graphs), both erasing nontrivial topology: the Gaussian limit washes out degree and communities, while the all-or-nothing densities ($\rho = 1$ or $0$) clash with the sparse and (dis)assortative patterns seen in real graphs. Thus, during reverse diffusion, the model must reinvent all higher-order statistics from white noise or a trivial edge set, yielding over-smoothed, overly dense samples or disconnected fragments that lack rich-club connectivity, disassortative hubs. In addition, existing studies predominantly inject isotropic Gaussian noise at each diffusion step, entirely oblivious to any directional preference. A handful of recent attempts replace this blind noise with locally directed perturbations, regulating the process through myopic neighborhood constraints (Yang et al., 2023). While these constraints guarantee single-step plausibility, they offer no mechanism to explicitly steer the terminal distribution, inevitably degrading graph-generation quality.

In this paper, we propose a Null-Model-Guided Graph Diffusion (NMG-GD), aiming to achieve high-quality graph generation and efficient sampling. In the forward diffusion process, we employ a stochastic differential equation (SDE) to characterize the evolution of graphs. By introducing directional diffusion, we progressively perturb the graph structure, ultimately converging to a n-order null model graph. It not only provides an intuitive theoretical foundation for the diffusion process, but also offers crucial topological information for the subsequent reverse generation process. In the reverse generation phase, we utilize reverse time SDEs to sample the original graph distribution from the null model graph. To this end, we leverage an efficient position-enhanced graph score network that extracts structural and positional cues from quantized graphs and fuses them with the continuous adjacency matrix, thereby effectively capturing the dynamic evolution of graph topology.

To summarize, our work makes the following contributions:

- Leveraging the structured randomness of null-model graphs to overcome the limitations of conventional diffusion priors serves. Null model graphs preserve critical topological properties (e.g., degree distributions, connectivity) while randomising nonessential structural features. We propose adopting null model graphs as the terminal state of the diffusion process, a design that explicitly embeds structural constraints into the diffusion trajectory.

- The design of directional noise enforces a global trajectory constraint, guiding the noise to progressively align with the structured randomness of the null model during diffusion—enabling more faithful capture of graph complexity and stochasticity. By prioritizing global optimality over local adjustments, our method overcomes the myopia of existing strategies, laying a rigorous foundation for high-fidelity graph generation.

- We conduct extensive experiments and the results demonstrate that the proposed NMG-GD outperforms state-of-the-art self-supervised methods and even supervised methods on 3 benchmark datasets. Additionally, we provide comprehensive ablation studies to gain a deeper understanding of the mechanisms underlying NMG-GD.

## 2 RELATED WORK

In addition to the graph generation approaches mentioned before, we summarize the notable existing literature on the construction of our framework.

### 2.1 DIFFUSION MODELS

Diffusion models have gained significant attention for their ability to generate high-quality data samples, particularly in the domain of image generation (Song & Ermon, 2020; Song et al., 2022; Kingma et al., 2023; Bao et al., 2022). The core concept of diffusion models is to progressively corrupt a data sample with noise through a forward diffusion process and then learn to reconstruct the original data through a reverse diffusion process (Sohl-Dickstein et al., 2015). This framework has been extended to various applications, including representation learning in computer vision. For example, Preechakul et al. (Preechakul et al., 2022) introduced Diff-AE, which integrates an encoder to capture high-level semantics and a conditional diffusion model that leverages these semantics as input conditions. Abstreiter et al. (Mittal et al., 2021) enhanced the denoising score matching framework to enable unsupervised representation learning.

## 2.2 GRAPH GENERATIVE MODELS

Graph generative models were originally proposed to generate diverse graphs based on the structural prior of target graph set (Müller et al., 1995). Subsequent deep-learning approaches—including autoregressive GraphRNN (You et al., 2018), VAE-based GraphVAE (Kipf & Welling, 2016), GRAN (Liao et al., 2020) and flow-based Graph Flows (Liu et al., 2019a)(Shi et al., 2020)—directly learn the distribution of observed graph collections (Goodfellow et al., 2014; Creswell et al., 2018; Gui et al., 2020; Li et al., 2018; Vahdat & Kautz, 2021). However, the limited capacity of these backbones yields unsatisfactory generation quality.

Consequently, several approaches have been proposed to develop diffusion models for the graph domain (Niu et al., 2020; Gnaneshwar et al., 2022; Jo et al., 2022; Vignac et al., 2023). BIGG (Dai et al., 2020) is the state-of-the-art autoregressive tree-based model which adopts a binary tree data structure to generate each edge and associates the set of edges with each node via a tree-structured autoregressive model. EDP-GNN (Niu et al., 2020) is a permutation invariance approach for graph generation via graph score matching and annealed Langevin dynamic sampling. GraphGDP (Huang et al., 2022) generates new graphs by reversing the process with a position-enhanced score network (PGSN) that equivariantly estimates scores from perturbed structure and position cues. **Pard** (Zhao et al., 2024) is a permutation-invariant autoregressive diffusion model that decomposes graph generation into a sequence of block-wise enlargements governed by a shared discrete denoising diffusion process. However, most existing models still adopt a Gaussian graph or a fully structured graph as the terminal distribution, overlooking the intrinsic structural properties of networks and leading to sub-optimal generative outcomes. Although some studies recognize the distinction between graphs and images and inject directional noise to guide the diffusion, they rely merely on local constraints(Yang et al., 2023), fail to control the final noise distribution, and thus cannot guarantee global statistical accuracy. In contrast, in our proposed NMG-GD framework, we leverage global constraints by setting the final distribution of the graph diffusion process to be the null model. Our method achieves a globally optimal solution rather than just a locally optimal one. Through this approach, we are able to generate high-quality graphs.

## 3 PRELIMINARIES

### 3.1 GRAPH DIFFUSION MODEL

Denoising diffusion model (Ho et al., 2020) is formulated as two Markov chains: a **forward** diffusion process that injects noise until the data distribution nearly collapses to an isotropic Gaussian, and a **reverse** denoising process that learns to iteratively restore the original data.

Formally, given an observed graph $G_0 \sim q(G_0)$, the forward chain progressively perturbs it into a sequence of noisy graphs $G_1, G_2, \ldots, G_T$ via a fixed variance schedule $\beta_t \in (0, 1)$:

$$q(G_t \mid G_{t-1}) = \mathcal{N}\Big(G_t; \sqrt{1 - \beta_t}\,G_{t-1},\, \beta_t \mathbf{I}\Big) \tag{1}$$

Under the Markov assumption, the joint distribution conditioned on $G_0$ factorizes as:

$$q(G_1, \ldots, G_T \mid G_0) = \prod_{t=1}^{T} q(G_t \mid G_{t-1}) \tag{2}$$

With $\alpha_t = 1 - \beta_t$ and $\bar{\alpha}_t = \prod_{s=1}^{t} \alpha_s$, any intermediate graph can be sampled in closed form:

$$G_t = \sqrt{\bar{\alpha}_t}\,G_0 + \sqrt{1 - \bar{\alpha}_t}\,\varepsilon, \quad \varepsilon \sim \mathcal{N}(\mathbf{0}, \mathbf{I}) \tag{3}$$

During training, a parameterized reverse transition $p_\theta(G_{t-1} \mid G_t)$ is optimised to minimize the variational lower bound (VLB) on the negative log-likelihood:

$$\mathcal{L}_{\text{VLB}} = \mathbb{E}_q \left[ \sum_{t=1}^{T} D_{\text{KL}}\big(q(G_{t-1} \mid G_t, G_0) \,\|\, p_\theta(G_{t-1} \mid G_t)\big) \right] \tag{4}$$

The discrete-time DDPM can be seamlessly transferred to a fully continuous formulation by treating the noise schedule. The forward chain converges to an Itô stochastic differential equation (SDE):

$$\mathrm{d}G_t = f(G_t, t)\,\mathrm{d}t + g(t)\,\mathrm{d}\mathbf{w}_t \tag{5}$$

where $\mathbf{w}_t$ denotes a standard Wiener process, and the drift coefficients $f(G_t, t)$ and diffusion co-efficients $g(t)$ are chosen to reproduce the noise schedule $\beta_t$ in the continuum limit. Under the variance-preserving (VP) prescription, for instance, one sets $f(t) = -\frac{1}{2}\beta(t)\,G_t$ and $g(t) = \sqrt{\beta(t)}$. The reverse denoising process similarly admits an SDE representation:

$$\mathrm{d}G_t = \Big[ f(G_t, t) - g^2(t)\,\nabla_G \log p_t(G) \Big]\mathrm{d}t + g(t)\,\mathrm{d}\bar{\mathbf{w}}_t \tag{6}$$

where $\bar{\mathbf{w}}_t$ is a reverse Brownian motion in time and $\nabla_G \log p_t(G)$ is the score function. The score is approximated by a neural network $\epsilon_\theta(G_t, t)$ trained via the continuous analog of the DDPM objective, resulting in a score-based generative model that continuously evolves from prior noise to data.

## 3.2 NULL MODEL

In network science, it is strictly defined as a randomized graph structure with explicit topological constraints (Váa & Mii, 2022); the aim is to preserve core attributes—degree sequence, component count, clustering, community partition—while maximally randomizing unconstrained wiring.The most basic null model is the configuration model: it freezes the entire degree sequence and rewires edges uniformly at random. This single constraint yields an ensemble whose deviations from the original reveal nonrandom edge preferences such as degree correlations or community structure.

Let the original network be an undirected simple graph $G = (V, E)$ with a degree sequence $\mathbf{k} = (k_1, \ldots, k_N)$, satisfying $\sum_{i=1}^{N} k_i = 2|M|$. Create $k_i$ stubs for each node $v_i$, yielding $2|M|$ stubs in total; uniformly pair stubs and link the corresponding nodes to obtain the configuration model null graph $G_0$. Repeat until no self-loops or multiple edges remain.

## 4 MODEL

In this section, we present the technical design of our proposed NMG-GD, accompanied by the overall model architecture depicted in Figure 1.

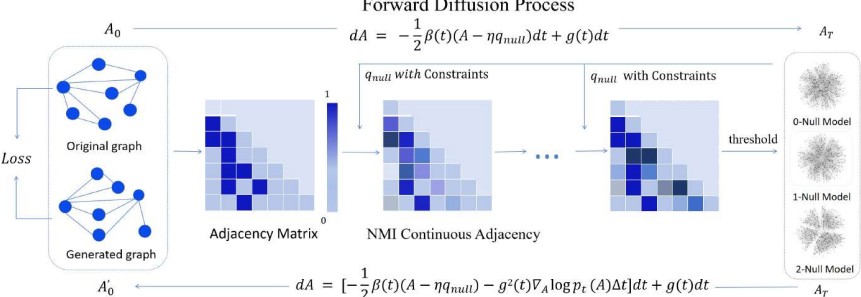

Figure 1: We inject constrained null-model noise into the adjacency matrix via a continuous-time SDE, yielding NMI adjacency matrices that terminate at the first-order null-model graph. The reverse SDE uses the score $\nabla_{A_t} \log p_t(A_t)$. Lower-triangle evolution is shown in the figure.

We perturbing the original graph toward a first-order null model graphs can be achieved through a continuous-time forward diffusion process described by an SDE. During the forward propagation process, constrained null model noise is added to the adjacency matrix of the graph to obtain NMI (Null Model Infused) Continuous Adjacency matrices, which gradually evolve into the terminal diffusion distribution-the first-order null model graph. We can convert this SDE with the score of the data distribution $\nabla_{A_t} \log p_t(A_t)$ at each time t.

Conventional graph diffusion models (Ho et al., 2020; Song et al., 2020) rely on sequentially adding isotropic Gaussian noise, driving the terminal state toward the standard normal distribution $\mathcal{N}(0, I)$. However, this paradigm is ill-suited for graph data owing to their pronounced structural heterogeneity, power-law degree distributions, community structure, and higher-order motifs. Two key limitations emerge: (1) Locality bias: Existing directed-diffusion approaches (Yang et al., 2023) adjust the noise direction only via batch-wise local moments, lacking an explicit global endpoint constraint. (2) Endpoint mismatch: Graph generation differs fundamentally from image synthesis; enforcing a fully structured graph as the terminal prior forcibly destroys salient topological properties such as degree sequences and clustering coefficients, leading to premature semantic collapse.

Building on this, We propose a **NMG-GD** framework that steers the diffusion process toward a null model distribution. Incorporating $q_{\text{null}}(A)$ lets the model retain prescribed structural invariants—e.g., degree sequences or edge-weight distributions—of the original graph, and we can switch among zeroth-order, first-order or second-order null models as the task demands. Adopting this null model as the terminal prior guides the diffusion to preserve key topological observables while preventing excessive randomization.

## 4.1 NULL-MODEL-GUIDED CONTINUOUS DIFFUSION MODEL

We explicitly revise the standard DDPM Markov kernel as a null-model-guided transition kernel:

$$A_t = \sqrt{\alpha_t}\, A_{t-1} + \sqrt{1 - \alpha_t}\, (\epsilon + \eta\, q_{\text{null}}(A))\,, \quad \varepsilon \sim \mathcal{N}(\mathbf{0}, \mathbf{I}) \tag{7}$$

This reformulation relocates the terminal distribution of the diffusion process from an isotropic Gaussian matrix to the manifold prescribed by a null model. As $t \to T$, the forward process drives $A_T$ toward the null-model distribution $\eta q_{\text{null}}(A)$ rather than the conventional $\mathcal{N}(0, I)$. Consequently, key graph statistics encoded in the null model—e.g., degree sequences, community strengths, or spectral radius—are preserved throughout denoising, precluding catastrophic loss of structural information. The term $\epsilon$ continues to supply isotropic Gaussian perturbations, whereas $\eta\, q_{\text{null}}(A)$ acts as an elastic restoring force that projects the sample toward the statistical manifold of the null model. In the limits $\eta = 0$ and $\eta = 1$ the process reduces, respectively, to the standard DDPM and to direct resampling from the null model.

Via a second-order Taylor expansion, we continuously extend the discrete formulation and thereby obtain its SDE representation.

We can decompose the given stochastic differential equation. Due to the limitation of space, the derivation of Equation (8) is provided in Appendix A.2.

$$dA_t = f(A, t)\, dt + g(t)\, dt \tag{8}$$

where, the **drift term** is

$$f(A, t) = -\frac{\beta(t)}{2}\big(A_t - \eta\, q_{\text{null}}\big) \tag{9}$$

the **diffusion coefficient** is

$$g(t) = \sqrt{\beta(t)}\varepsilon'(t), \varepsilon'(t) \sim N(\frac{1}{2}\eta\, q_{\text{null}}, 1) \tag{10}$$

In the limit $T \to \infty$ the process converges to the stationary distribution $A_T \sim q_{\text{null}}(A)$, ensuring global consistency and a physically interpretable terminal state.

## 4.2 NULL-MODEL-GUIDED REVERSE PROPAGATION

Building upon the null-model-guided forward process in Equation (8), the corresponding reverse time SDE takes the explicit form. The derivation of Equation (11) is provided in Appendix A.3.

$$dA_t = \Big[f(A, t) - g^2(t)\, \nabla_A \log p_t(A)\Delta t\Big]dt + g(t)\varepsilon'(t)\, dt \tag{11}$$

where $\nabla_{A_t} \log p_t(A_t)$ is the **null-model-guided score function**, which refines local structure and effects controlled convergence from chaotic noise to the true graph distribution.

We directly use the Position-Enhanced Graph Score Network(PGSN) (Huang et al., 2022) as our score estimator

$$s_\theta(A_t, \bar{A}_t, t) \approx \nabla_{A_t} \log p_t(A_t) \tag{12}$$

where $\bar{A}_t$ is a quantized graph obtained by thresholding or reparameterizing the continuous adjacency matrix, thereby reconciling the inherent tension between discrete graph data and the continuous SDE framework. PGSN is permutation-equivariant, guaranteeing the invariance required for graph isomorphism.

PGSN takes as input the continuous perturbed adjacency matrix $A_t$ and internally computes its discrete counterpart $\bar{A}_t$ via thresholding. Node degrees are encoded into one-hot vectors to capture low-order statistics. For each node, an $r$-step random-walk landing-probability vector is computed, resulting in the Random Walk Structural Encoding (RWSE) that reflects global reachability. These features are projected and processed through a series of message-passing Graph Neural Network (GNN) layers. At each layer, edge embeddings are refined by fusing incident node representations with the corresponding RWSE and Shortest-Path-Distance (SPD) encodings.It outputs an edge-level score $s_\theta(A_t, \bar{A}_t, t)$ that is permutation-equivariant.

## 4.3 LOSS FUNCTION

The loss function is a critical component in the training process, as it directly influences the model's ability to learn from the data and generalize to unseen examples.

Since the graphs are undirected, we only manipulate the lower triangular part of the adjacency matrix and then perform symmetrization to obtain the complete symmetric adjacency matrix.

According to the definition of the forward diffusion process (refer to Equation (8)), combined with the property of affine drift coefficients, the perturbation kernel $p_{0t}(\mathbf{A}_t|\mathbf{A}_0)$ (i.e., the marginal distribution of graph $\mathbf{A}_t$ at time $t$ with respect to the initial graph $\mathbf{A}_0$) for the null model-guided diffusion is expressed as follows:

First, the basic form of the perturbation kernel without the null model is provided:

$$p_{0t}(\mathbf{A}_t|\mathbf{A}_0) = \mathcal{N}\left(\mathbf{A}_t; \mathbf{A}_0 e^{-\frac{1}{2}\int_0^t \beta(s)\mathrm{d}s}, \mathbf{I} - \mathbf{I}e^{-\int_0^t \beta(s)\mathrm{d}s}\right) \tag{13}$$

The perturbation kernel with the null model incorporated is

$$p_{0t}(\mathbf{A}_t|\mathbf{A}_0) = \mathcal{N}\left(\mathbf{A}_t; \ \mathbf{A}_0\, e^{-\frac{1}{2}\int_0^t \beta(s)\mathrm{d}s} + \eta\big(\mathbf{I} - e^{-\frac{1}{2}\int_0^t \beta(s)\mathrm{d}s}\big) q_{\text{null}}(\mathbf{A}_0), \mathbf{I} - \mathbf{I}\, e^{-\int_0^t \beta(s)\mathrm{d}s}\right) \tag{14}$$

Here, $q_{\text{null}}(\mathbf{A}_0)$ denotes the null model constructed based on the initial graph $\mathbf{A}_0$ (e.g., the configuration model preserving the node degree sequence), and $\eta \in [0,1]$ is the null model weight parameter, which adjusts the influence of the null model on the mean of the diffusion process. The derivation of Equation (14) is provided in Appendix A.4.

Using the aforementioned perturbation kernel, we can perturb the initial graph, without the need to execute the forward diffusion process step-by-step. While the forward diffusion process directly operates on the distribution of continuous-domain adjacency matrices, it also implicitly defines a corresponding transformation on the discrete-domain graph distribution: by simply quantizing the continuously sampled $\mathbf{A}_t$ (setting a discretization threshold of 0.3 for edge values), we can convert it into a discrete graph $\bar{\mathbf{A}}_t$. As diffusion proceeds, the structural signals specific to the original graph are gradually erased, yet the core topological constraints encoded in $q_{\text{null}}(\mathbf{A}_0)$—such as the node degree sequence -level statistical properties—remain preserved.

Finally, the training objective of the model is defined as follows.

$$\min_\theta \ \mathbb{E}_t\Big\{\lambda(t)\,\mathbb{E}_{\mathbf{A}_0}\,\mathbb{E}_{\mathbf{A}_t|\mathbf{A}_0}\Big[\,\big\|\mathbf{s}_\theta(\mathbf{A}_t, \bar{\mathbf{A}}_t, t) - \nabla_{\mathbf{A}} \log p_{0t}(\mathbf{A}_t|\mathbf{A}_0)\big\|_2^2\,\Big]\Big\} \tag{15}$$

Here, $\theta$ denotes the parameters of the score network $\mathbf{s}_\theta(\cdot)$, $\nabla_{\mathbf{A}} \log p_{0t}(\mathbf{A}_t|\mathbf{A}_0)$ is the gradient of the log-probability of the perturbation kernel with respect to $\mathbf{A}_t$ (i.e., the analytical score function),

and $\lambda(t)$ is a time-varying weight coefficient used to balance the training contribution of different diffusion time steps.

# 5 EXPERIMENTS

In this section, we empirically demonstrate the power of the proposed NMG-GD in the task of graph generation.

## 5.1 DATASETS

We evaluate our graph generative model across four diverse graph datasets, which differ in terms of graph scale and attributes.

1) ***Community-small:*** Comprises 100 community graphs where $12 \leq |V| \leq 20$. These graphs are composed of two equally sized communities, each generated via the Erdős-Rényi model (E-R) (Erd6s & Rényi, 1960) with $p = 0.7$.

2) ***Ego-small:*** Involves 200 one-hop ego graphs where $4 \leq |V| \leq 18$, sourced from the Citeseer network (Sen et al., 2008). Here, nodes correspond to documents and edges signify citation relationships.

3) ***Enzymes:*** Includes 563 protein graphs where $10 \leq |V| \leq 125$, selected from the BRENDA database (Schomburg et al., 2004).

We further partition the datasets into training and test sets at a ratio of 8:2. The validation set is derived from the initial 20% of the training graphs. When assessing model performance on Community-small and Ego-small, we produce 1024 graph samples in accordance with (Liu et al., 2019a) and (Niu et al., 2020) to achieve more robust evaluation outcomes for small graphs. For the Enzymes datasets, we generate a number of graphs equivalent to the size of the test set.

## 5.2 EVALUATION METRICS

Assessing and comparing graph generative models presents a significant challenge. We employ two types of metrics to conduct a thorough evaluation of the quality of graph generation.

1) *Classical Structure Metrics:* The widely-used evaluation metrics are based on **Maximum Mean Discrepancy (MMD) measures** to assess the distance between the distributions of the generated graph set $\mathbb{S}_g$ and the test set $\mathbb{S}_t$ (Liu et al., 2019a; You et al., 2018; Liao et al., 2020; Dai et al., 2020; Chen et al., 2021; Niu et al., 2020). We employ three graph-level structure descriptor functions, which are described in (You et al., 2018; Liao et al., 2020), including the **degree distribution**, the **clustering coefficient distribution** and the **Laplacian spectrum histograms**. We adopt the Total-Variation (TV) kernel as an efficient and valid Mercer-like kernel.This formulation replaces the conventional Euclidean distance with an anisotropic measure that penalizes jumps along coordinate directions, thereby retaining sharp transitions while still furnishing a smooth similarity score.

2) *Neural-network-based Metrics:* Thompson *et al.* (Thompson et al., 2022) introduce several random GIN-based metrics for graph generative model evaluation, as the pre-existing structure metrics fail to capture the diversity of graph samples. The graph representations are extracted by random-initialized GIN (Xu et al., 2018), where **MMD** (*i.e.*, MMD computed with the TV kernel), **F1 PR** (*i.e.*, the harmonic mean of improved precision and recall) and **F1 DC** (*i.e.*, the harmonic mean of density and coverage) are built.

For Classical Structural Metrics, the smaller the value, the better the performance of the generated graph. The specific metrics include: degree distribution (Deg.), clustering coefficient distribution (Clus.), and spectrum of graph Laplacian (Spec.). Additionally, lower average values of three MMD metrics (Avg.) indicate closer alignment between generated and reference distributions, thus better performance. For Neural-Network-Based Measures, MMD (the smaller the value, the better), F1 PR and F1 DC (the closer the value is to 1, the better) are used to evaluate the performance. The top cell in each column is marked in bold according to its rank.

## 5.3 BASELINES

We compare the performance of our models against other graph generative models including ER (Erdös & Rényi, 2006), VGAE (Kipf & Welling, 2016), GraphRNN (You et al., 2018), GRAN (Liao et al., 2020), GraphRNN-U (the random-order GraphRNN) and GRAN-U (the random-order GRAN), EDP-GNN (Niu et al., 2020), BIGG (Dai et al., 2020), GraphGDP (Huang et al., 2022) and Pard (Zhao et al., 2024).

## 5.4 GRAPH GENERATION QUALITY

In the forward SDE, we fix $\bar{\beta}_{\min} = 0.1$, select $\bar{\beta}_{\max} \in \{5, 10, 20\}$, and set $\eta = 0.1$. All samples are subsequently refined via Langevin-corrected dynamics.

Table 1: Comparison of performance of graph generation models with classical structural metrics.

| | Community-small $\|V\|_{\max} = 20, \|E\|_{\max} = 62$ $\|V\|_{\text{avg}} \approx 15, \|E\|_{\text{avg}} \approx 36$ | | | | Ego-small $\|V\|_{\max} = 17, \|E\|_{\max} = 66$ $\|V\|_{\text{avg}} \approx 6, \|E\|_{\text{avg}} \approx 9$ | | | | Enzymes $\|V\|_{\max} = 125, \|E\|_{\max} = 149$ $\|V\|_{\text{avg}} \approx 33, \|E\|_{\text{avg}} \approx 63$ | | | |
|---|---|---|---|---|---|---|---|---|---|---|---|---|
| | Deg. | Clus. | Spec. | Avg. | Deg. | Clus. | Spec. | Avg. | Deg. | Clus. | Spec. | Avg. |
| GraphRNN | 0.106 | 0.115 | 0.091 | 0.104 | 0.155 | 0.229 | 0.167 | 0.184 | 0.397 | 0.302 | 0.260 | 0.320 |
| GRAN | 0.125 | 0.164 | 0.111 | 0.133 | 0.096 | 0.072 | 0.095 | 0.088 | 0.215 | 0.147 | 0.034 | 0.132 |
| BIGG | 0.041 | 0.073 | 0.050 | 0.055 | 0.024 | 0.029 | 0.028 | 0.027 | 0.020 | 0.019 | 0.019 | 0.019 |
| ER | 0.300 | 0.239 | 0.100 | 0.213 | 0.200 | 0.094 | 0.361 | 0.218 | 0.844 | 0.381 | 0.104 | 0.443 |
| VGAE | 0.391 | 0.257 | 0.095 | 0.248 | 0.146 | 0.046 | 0.249 | 0.147 | 0.811 | 0.514 | 0.153 | 0.493 |
| GraphRNN-U | 0.410 | 0.297 | 0.103 | 0.270 | 0.471 | 0.416 | 0.398 | 0.429 | 0.932 | 1.000 | 0.367 | 0.766 |
| GRAN-U | 0.106 | 0.127 | 0.083 | 0.106 | 0.155 | 0.229 | 0.167 | 0.184 | 0.343 | 0.122 | 0.041 | 0.169 |
| EDP-GNN | 0.100 | 0.140 | 0.085 | 0.108 | 0.026 | 0.032 | 0.037 | 0.032 | 0.120 | 0.644 | 0.070 | 0.278 |
| GraphGDP | 0.039 | 0.074 | 0.052 | 0.055 | 0.023 | 0.029 | 0.030 | 0.027 | 0.023 | **0.025** | 0.019 | 0.022 |
| Pard | **0.023** | 0.071 | 0.186 | 0.093 | 0.037 | 0.091 | 0.032 | 0.053 | **0.008** | 0.026 | **0.013** | **0.016** |
| NMG-GD | 0.037 | **0.047** | **0.043** | **0.042** | **0.021** | **0.024** | **0.021** | **0.022** | 0.023 | 0.029 | 0.019 | 0.023 |

Table 2: Comparison of performance of graph generation models with neural-network-based measures.

| | Community-small | | | Enzymes | | |
|---|---|---|---|---|---|---|
| | MMD ($\downarrow$) | F1 PR ($\uparrow$) | F1 DC ($\uparrow$) | MMD ($\downarrow$) | F1 PR ($\uparrow$) | F1 DC ($\uparrow$) |
| GraphRNN | $0.353 \pm 0.088$ | $0.252 \pm 0.183$ | $0.407 \pm 0.171$ | $1.495 \pm 0.037$ | $0.000 \pm 0.000$ | $0.000 \pm 0.000$ |
| GRAN | $0.196 \pm 0.014$ | $0.824 \pm 0.141$ | $0.793 \pm 0.099$ | $0.069 \pm 0.008$ | $0.915 \pm 0.035$ | $0.738 \pm 0.027$ |
| BIGG | $0.052 \pm 0.003$ | $0.135 \pm 0.087$ | $1.048 \pm 0.035$ | $0.019 \pm 0.000$ | $0.964 \pm 0.008$ | $0.966 \pm 0.012$ |
| ER | $0.278 \pm 0.046$ | $0.363 \pm 0.201$ | $0.335 \pm 0.096$ | $0.808 \pm 0.065$ | $0.046 \pm 0.030$ | $0.019 \pm 0.005$ |
| VGAE | $0.360 \pm 0.065$ | $0.292 \pm 0.165$ | $0.292 \pm 0.113$ | $0.716 \pm 0.033$ | $0.012 \pm 0.016$ | $0.002 \pm 0.003$ |
| GraphRNN-U | $0.970 \pm 0.113$ | $0.066 \pm 0.043$ | $0.079 \pm 0.003$ | $1.263 \pm 0.177$ | $0.000 \pm 0.000$ | $0.000 \pm 0.000$ |
| GRAN-U | $0.164 \pm 0.016$ | $0.859 \pm 0.082$ | $0.888 \pm 0.053$ | $0.242 \pm 0.033$ | $0.671 \pm 0.056$ | $0.364 \pm 0.024$ |
| EDP-GNN | $0.125 \pm 0.004$ | $0.913 \pm 0.108$ | $0.977 \pm 0.044$ | $0.119 \pm 0.010$ | $0.954 \pm 0.012$ | $0.846 \pm 0.020$ |
| GraphGDP | $0.066 \pm 0.012$ | $0.656 \pm 0.138$ | $1.042 \pm 0.014$ | $0.026 \pm 0.001$ | $0.974 \pm 0.005$ | $0.932 \pm 0.015$ |
| Pard | $0.165 \pm 0.021$ | **$0.790 \pm 0.030$** | $0.475 \pm 0.073$ | $0.053 \pm 0.005$ | $0.952 \pm 0.023$ | $0.883 \pm 0.049$ |
| NMG-GD | **$0.065 \pm 0.011$** | $0.692 \pm 0.115$ | **$1.029 \pm 0.054$** | **$0.020 \pm 0.002$** | **$0.979 \pm 0.006$** | **$0.933 \pm 0.014$** |

We summarize the key observations after evaluating the proposed NMG-GD for graph generation. (1) Among all competitive baseline models, our method achieves a marked improvement in both convergence stability and generation fidelity. (2) Compared with conventional methods such as GraphRNN, graph diffusion models demonstrate superior performance across both classical and neural-network-based metrics. (3) Relative to existing graph diffusion models, the proposed NMG-GD model achieves remarkable performance gains by incorporating direction noise with global constraints and by refining the terminal state.

Figure 2 illustrates the reverse generation process of two exemplar instances of NMG-GD. In the early stages of generation, the model produces a graph that approximates the first-order null model of the original. As the number of iterations increases, the generated graph retains progressively more structural characteristics (e.g., community structure) and thereby closely approximates the distribution of the original graph.

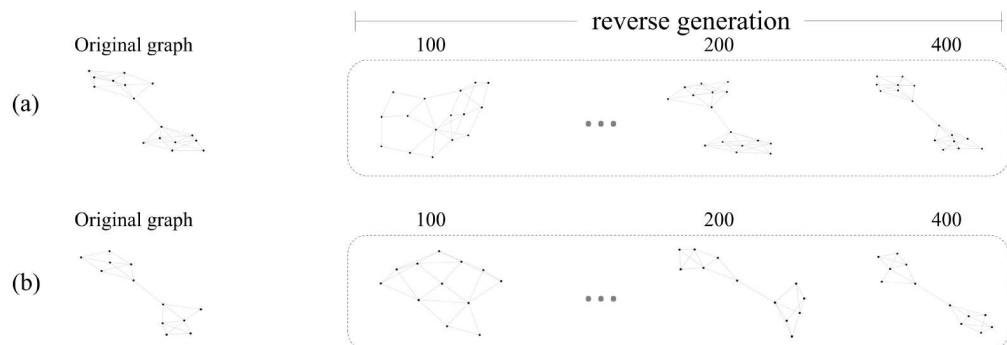

Figure 2: Graph visualization of different steps in the generative diffusion processes on Community-small (a-b).

## 5.5 ABLATION STUDY AND PARAMETER SENSITIVITY ANALYSIS

To validate the effectiveness of the design for NMG-GD, we compare it with its variants using generative diffusion processes on the Community-small dataset. All models are trained with a unified configuration of 64 hidden dimensions and 4 million training steps.

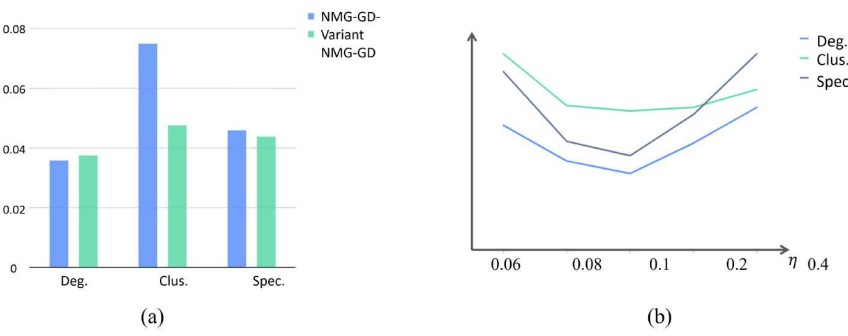

Figure 3: (a) Ablation Study; (b) Parameter Sensitivity Analysis w.r.t. $\eta$.

The dynamics of NMG-GD are described by Equation (8). The variant of NMG-GD evolves according to

$$dA_t = -\frac{\beta(t)}{2}\big(A_t - \eta\, q_{\text{null}}\big)\, dt + \sqrt{\beta(t)}\, dw \tag{16}$$

As evidenced by the results, NMG-GD consistently surpasses this variant across all metrics, thereby confirming the effectiveness of the proposed design.

To identify the optimal parameter $\eta$, we conducted a parameter sensitivity analysis experiment on the Community-small dataset over $3 \times 10^6$ training iterations. The outcomes are summarised in the accompanying Figure 3(b). We can find that the optimal value of $\eta$ varies under different evaluation metrics.

## 6 CONCLUSIONS

In this work, we elevate the diffusion terminus from Gaussian graph or fully structural graph to a first-order null-model graph, substituting global statistical constraints for local directional perturbations. The resulting null-model-guided stochastic differential equation (SDE) yields a terminal state that retains essential structural information. This paradigm shift endows continuous-time graph diffusion with both theoretical transparency and enhanced sample quality. Extending the framework to attributed or temporal graphs, constitutes an immediate direction for future research.

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

# A APPENDIX

## A.1 DDPM FORWARD KERNEL

In diffusion models, given a sample $A_0 \sim q_(A)$, the forward process progressively adds Gaussian noise $T$ times, yielding $x_1, x_2, \ldots, x_T$. The magnitude of each perturbation is governed by a sequence of variance hyper-parameters $\beta_t$. Because each time step depends only on its immediate predecessor, the chain is Markovian.

Let $\beta_t$ denote the variance scheduled at step $t$. As $t$ increases, $x_t$ converges toward pure noise; in the limit $t \to \infty$ it becomes standard Gaussian. The conditional transition can therefore be written

$$q(A_t \mid A_{t-1}) = \mathcal{N}\big(A_t; \sqrt{1 - \beta_t}\, A_{t-1}, \beta_t \mathbf{I}\big), \tag{17}$$

After reparameterized sampling, $q(A_t \mid A_{t-1}) = \mathcal{N}\big(A_t; \sqrt{1 - \beta_t} A_{t-1}, \beta_t \mathbf{I}\big)$ can be written as:

$$A_t = \sqrt{\alpha_t}\, A_{t-1} + \sqrt{\beta_t}\, \varepsilon_t, \qquad \varepsilon \sim \mathcal{N}(\mathbf{0}, \mathbf{I}). \tag{18}$$

where $\alpha_t = 1 - \beta_t$ with $\beta_t \in (\mathbf{0}, \mathbf{I})$.

## A.2 DERIVATION OF EQUATION (8)

We wish the terminal distribution to be a null model graph; consequently, the discrete-time update is

$$A_t = \sqrt{\alpha_t}\, A_{t-1} + \sqrt{1 - \alpha_t}\, \big(\varepsilon + \eta\, q_{\text{null}}(A)\big), \quad \varepsilon \sim \mathcal{N}(\mathbf{0}, \mathbf{I}) \tag{19}$$

To move from discrete steps to a continuous-time formulation, we introduce the continuous limit.

$$A_t = A(t + \tfrac{1}{T}), \quad A_{t-1} = A(t), \quad \varepsilon_{t-1} = \varepsilon(t), \quad \beta_t = \tfrac{1}{T}\beta(t + \tfrac{1}{T}) \tag{20}$$

so that the single-step update becomes

$$A(t + \Delta t) = \sqrt{1 - \beta(t + \Delta t)\Delta t}\, A(t) + \sqrt{\beta(t + \Delta t)\Delta t^2}\, \big(\varepsilon(t) + \eta\, q_{\text{null}}(A)\big) \tag{21}$$

Using the Taylor expansion $\sqrt{1 - x} \approx 1 - \frac{x}{2}$:

$$A(t + \Delta t) \approx \Big[1 - \frac{\beta(t + \Delta t)}{2}\Delta t\Big] A(t) + \sqrt{\beta(t + \Delta t)\Delta t^2}\, \big(\varepsilon(t) + \eta\, q_{\text{null}}(A)\big) \tag{22}$$

$$\begin{aligned}
A(t + \Delta t) - A(t) \\
\approx -\frac{\beta(t + \Delta t)}{2}\Delta t\, A(t) + \sqrt{\beta(t + \Delta t)\Delta t^2}\, \varepsilon(t) + \sqrt{\beta(t + \Delta t)\Delta t^2}\, \eta\, q_{\text{null}}(A) \\
\approx -\frac{\beta(t)\Delta t}{2} A(t) + \sqrt{\beta(t)\Delta t^2}\, \varepsilon(t) + \sqrt{\beta(t)\Delta t^2}\, \eta\, q_{\text{null}}(A)
\end{aligned} \tag{23}$$

$$dA(t) = -\frac{\beta(t)}{2} A(t)\, dt + \sqrt{\beta(t)}\, \varepsilon(t)\, dt + \sqrt{\beta(t)}\, \eta\, q_{\text{null}}(A)\, dt \tag{24}$$

Collecting drift terms gives the final guided forward SDE

$$dA(t) = -\frac{\beta(t)}{2}\big(A(t) - \eta\, q_{\text{null}}(A)\big)dt + \sqrt{\beta(t)}\, \big(\varepsilon(t) + \tfrac{1}{2}\eta\, q_{\text{null}}(A)\big)dt \tag{25}$$

Let $\varepsilon'(t) = \varepsilon(t) + \tfrac{1}{2}\eta\, q_{\text{null}}(A)$.

Since $\varepsilon(t) \sim \mathcal{N}(\mathbf{0}, \mathbf{I})$, the combined noise term is distributed as

$$\varepsilon'(t) \sim \mathcal{N}\left(\frac{1}{2}\eta\, q_{\text{null}}(A), \mathbf{I}\right) \tag{26}$$

Introducing the shorthand $A_t$ for the guided process we obtain

$$\mathrm{d}A(t) = -\frac{\beta(t)}{2}\big(A(t) - \eta\, q_{\text{null}}(A)\big)\mathrm{d}t + \sqrt{\beta(t)}\,\varepsilon'(t)\mathrm{d}t, \quad \varepsilon'(t) \sim \mathcal{N}\left(\frac{1}{2}\eta q_{\text{null}}, \mathbf{I}\right) \tag{27}$$

We hereby redefine $f(A,t)$ and $g(t)$ as follows:

$$f(A,t) = -\frac{\beta(t)}{2}\big(A(t) - \eta\, q_{\text{null}}(A)\big), \tag{28}$$

$$g(t) = \sqrt{\beta(t)}\,\varepsilon'(t)$$

### A.3 DERIVATION OF EQUATION (11)

We now derive the reverse transition $(A_t \mid A_{t+\Delta t})$. Starting from the Bayes decomposition

$$
\begin{aligned}
q(A_{t-1} \mid A_t, A_0) &= \frac{q(A_{t-1}, A_t, A_0)}{q(A_t, A_0)} \\
&= \frac{q(A_t \mid A_{t-1})\, q(A_{t-1} \mid A_0)\, q(A_0)}{q(A_t \mid A_0)\, q(A_0)} \\
&= \frac{q(A_t \mid A_{t-1})\, q(A_{t-1} \mid A_0)}{q(A_t \mid A_0)}
\end{aligned}
\tag{29}
$$

we write the continuous analogue as

$$\frac{q(A_{t+\Delta t} \mid A_t)\, q(A_t)}{q(A_{t+\Delta t})} \tag{30}$$

The explicit expressions of the factors in Equation (29) are presented below.

$$q(A_{t+\Delta t} \mid A_t) = \frac{1}{\sqrt{2\pi}\, g_t \Delta t}\exp\left(-\frac{A_{t+\Delta t} - A_t - f(A,t)\Delta t - \frac{1}{2}\eta\, q_{\text{null}}(A)g_t^2(\Delta t)^2}{2g_t^2\Delta t^2}\right). \tag{31}$$

$$\frac{q(A_t)}{q(A_{t+\Delta t})} = \exp\big(\log q(A_t) - \log q(A_{t+\Delta t})\big) \tag{32}$$

Performing a Taylor expansion on $\log q(A_{t+\Delta t})$:

$$\log p(A_{t+\Delta t}) = \log p(A_t) + (A_{t+\Delta t} - A_t)\nabla_A \log p(A_{t+\Delta t}) + (t + \Delta t - t)\nabla_A^2 \log p(A_{t+\Delta t}) + \cdots \tag{33}$$

Hence Equation (32) becomes

$$\exp\big((-A_{t+\Delta t} + A_t)\nabla_A \log p(A_{t+\Delta t})\big) \tag{34}$$

Collecting exponents and completing the square gives the Gaussian reverse kernel

$$
\begin{aligned}
Equation\ (30) = {}&\frac{1}{\sqrt{2\pi}\, g_t \Delta t}\exp\left(-\frac{\big(A_{t+\Delta t} - A_t - f(A,t)\Delta t - \frac{1}{2}\eta\, q_{\text{null}}(A)g_t\Delta t\big)^2}{2g_t^2\Delta t^2}\right) \\
&\times \exp\left(-\frac{(A_{t+\Delta t} - A_t)\nabla_A \log p(A_{t+\Delta t}) \cdot 2g_t^2\Delta t^2}{2g_t^2\Delta t^2}\right)
\end{aligned}
\tag{35}
$$

We now analyze the quadratic form in the exponential factor:

$$\mathcal{Q}_t = -\frac{1}{2g_t^2\Delta t^2}\left[A_t - \left(A_{t+\Delta t} - \left(f(A,t) + \tfrac{1}{2}\eta\, q_{\text{null}}(A)g_t - g_t^2\nabla_A \log p(A_{t+\Delta t})\Delta t\right)\Delta t\right)\right]^2 \quad (36)$$

to highlight the mean and variance of the reverse transition. The resulting update is

$$A_t = A_{t+\Delta t} - \left(f(A,t) + \tfrac{1}{2}\eta\, q_{\text{null}}(A)g_t - g_t^2\nabla_A \log p(A_{t+\Delta t})\Delta t\right)\Delta t + g_{t+\Delta t}\Delta t\,\varepsilon \quad (37)$$

or equivalently

$$A_t - A_{t+\Delta t} = \left(f(A,t) + \tfrac{1}{2}\eta\, q_{\text{null}}(A)g_t - g_t^2\nabla_A \log p(A_{t+\Delta t})\Delta t\right)\Delta t - g_{t+\Delta t}\Delta t\,\varepsilon \quad (38)$$

Re-expressed as a forward difference ($\Delta t > 0$) we obtain

$$A_{t+\Delta t} - A_t = \left(f(A,t) + \tfrac{1}{2}\eta\, q_{\text{null}}(A)g_t - g_t^2\nabla_A \log p(A_{t+\Delta t})\Delta t\right)\Delta t + g_{t+\Delta t}\Delta t\,\varepsilon. \quad (39)$$

Taking the limit $\Delta t \to dt$ gives the reverse time SDE

$$dA_t = \left[f(A,t) - g^2(t)\,\nabla_A\log p_t(A)\Delta t\right]dt + g(t)\varepsilon'(t)\,dt, \varepsilon' \sim \mathcal{N}\left(\tfrac{1}{2}\eta\, q_{\text{null}}(A), 1\right) \quad (40)$$

## A.4 DERIVATION OF EQUATION (14)

The basic form of the perturbation kernel without the null model is:

$$p_{0t}(\mathbf{A}_t|\mathbf{A}_0) = \mathcal{N}\left(\mathbf{A}_t; \mathbf{A}_0 e^{-\frac{1}{2}\int_0^t \beta(s)\mathrm{d}s}, \mathbf{I} - \mathbf{I}e^{-\int_0^t \beta(s)\mathrm{d}s}\right) \quad (41)$$

From Equation (8), we proceed to derive its mean and variance.

The mean $\mathbb{E}[A(t)]$ satisfies the following deterministic differential equation:

$$\frac{d\mathbb{E}[A(t)]}{dt} = -\frac{\beta(t)}{2}\left(\mathbb{E}[A(t)] - \eta q_{\text{null}}(A)\right) \quad (42)$$

This is a first-order linear ordinary differential equation, which can be written in the standard form:

$$\frac{dy}{dt} + P(t)y = Q(t) \quad (43)$$

where $y = \mathbb{E}[A(t)]$, $P(t) = \frac{\beta(t)}{2}$, and $Q(t) = \frac{\beta(t)}{2}\eta q_{\text{null}}(A)$.

According to the solution method for first-order linear ordinary differential equations, the solution is given by:

$$y(t) = e^{-\int P(t)dt}\left(\int Q(t)e^{\int P(t)dt}dt + C\right) \quad (44)$$

Substituting $P(t)$ and $Q(t)$ into the equation, we obtain:

$$\mathbb{E}[A(t)] = e^{-\frac{1}{2}\int_0^t \beta(s)ds}\left(\int_0^t \frac{\beta(s)}{2}\eta q_{\text{null}}(A)e^{\frac{1}{2}\int_0^s \beta(u)du}ds + C\right) \quad (45)$$

Assuming the initial condition $\mathbb{E}[A(0)] = A_0$, we have $C = A_0 - \eta q_{\text{null}}(A)$. Therefore:

$$\mathbb{E}[A(t)] = \eta q_{\text{null}}(A) + (A_0 - \eta q_{\text{null}}(A))\, e^{-\frac{1}{2}\int_0^t \beta(s)ds} \quad (46)$$

Compared to the VP-SDE, we have not altered the variance, hence it remains as:

$$\mathbf{I} - \mathbf{I}e^{-\int_0^t \beta(s)ds} \quad (47)$$

Therefore, the mean and variance are as follows:

$$\boldsymbol{\mu}_t = \mathbf{A}_0 e^{-\frac{1}{2}\int_0^t \beta(s)\mathrm{d}s} + \eta\left(\mathbf{I} - e^{-\frac{1}{2}\int_0^t \beta(s)\mathrm{d}s}\right)q_{\text{null}}(\mathbf{A}_0), \qquad \boldsymbol{\Sigma}_t = \mathbf{I} - \mathbf{I}e^{-\int_0^t \beta(s)\mathrm{d}s}. \quad (48)$$

The perturbation kernel with the null model incorporated is:

$$p_{0t}(\mathbf{A}_t|\mathbf{A}_0) = \mathcal{N}\left(\mathbf{A}_t;\ \mathbf{A}_0\, e^{-\frac{1}{2}\int_0^t \beta(s)\mathrm{d}s} + \eta\left(\mathbf{I} - e^{-\frac{1}{2}\int_0^t \beta(s)\mathrm{d}s}\right)q_{\text{null}}(\mathbf{A}_0), \mathbf{I} - \mathbf{I}e^{-\int_0^t \beta(s)\mathrm{d}s}\right) \quad (49)$$