# OpenReview forum: "Exploring Effective Terminal State: A Null-Model-Guided Graph Diffusion Model"
_ICLR.cc/2026/Conference — Submitted to ICLR 2026_

### Official Review · Reviewer_dfPc · 2025-10-28

**Soundness:** 2
**Presentation:** 3
**Contribution:** 3
**Rating:** 4
**Confidence:** 3

**Summary:**

This paper proposes a novel graph diffusion model, NMG-GD, that addresses that existing graph diffusion methods use an unstructured or overly simplistic terminal state, which erases crucial topological properties of the original graph, forcing the model to reinvent complex statistics during the reverse process. NMG-GD guides the forward diffusion process to terminate at a first-order null model graph, preserving essential global statistics.  Extensive experiments on synthetic and real-world biological networks demonstrate that NMG-GD achieves competitive performance across multiple structural and neural network-based metrics.

**Strengths:**

1. This paper conducts comprehensive experiments on multiple datasets, showing that NMG-GD consistently outperforms a wide range of baselines, including recent diffusion models, across both classical structural metrics and modern neural metrics.

2. This paper conducts an ablation study comparing a variant without the full noise design and a parameter sensitivity analysis, helping to understand the contribution of its components.

3. The derivation of the null-model-guided SDE for both the forward and reverse processes is detailed and appears sound.

**Weaknesses:**

1. The paper emphasizes the use of directional noise as a key contribution over prior work. However, the connection between the proposed method and the concept of directionality is not fully clarified. The introduced noise $ε^'$ remains Gaussian, albeit with a shifted mean. It is better to discuss how this constitutes directional guidance rather than isotropic noise.

2. The paper does not explain why the proposed NMG-GD works so well. Is the primary benefit simply that it provides a better-initialized starting point for the reverse process? Or does it fundamentally reshape the loss landscape of the score function, making it easier to learn? It is better to conduct a deeper analysis, for example, by visualizing the diffusion trajectory.

3. The paper does not discuss the computational complexity or the practical overhead of NMG-GD compared to other baselines.

**Questions:**

1. Why is the discretization threshold set at 0.3 for converting the continuous adjacency matrix into a discrete graph? What was the rationale behind selecting this specific value? Was this threshold fine-tuned, and how sensitive is the model's performance to it?

2. NMG-GD uses a continuous adjacency matrix and applies a fixed threshold of 0.3 for discretization. What are the potential effects of exploring other discretization strategies, such as sampling edges from a Bernoulli distribution that is parameterized by the continuous values? Additionally, how does the choice of the discretization method influence the quality of the final discrete graph?

---

> ### Author Response · Authors · 2025-12-03
> **#17202-We have provided detailed responses regarding the structure-preserving motivation of the weakly directional noise, the role of NMG-GD, the computational overhead, the fair use of the 0.3 threshold, and the simplicity and efficiency of the continuous-adjacency-matrix-plus-fixed-threshold strategy.**
>
> Thank you for your meticulous review and valuable feedback. We have provided detailed responses based on your suggestions below.
>
> Weakness:
>
> 1. The directional noise proposed in this paper is only weakly directional; its purpose is to add noise while preserving the original structural properties, rather than to impose strict constraints on the directionality of the noise itself.
>
> 2. The effectiveness of NMG-GD stems mainly from providing a generation path for the reverse process, enabling better exploitation of structural features.
>
> 3. Due to space limitations, the computational overhead of NMG-GD will be reported in the revised version.
>
> Question:
>
> 1. Following most existing models, we set the discretization threshold to 0.3 for fairness.
>
> 2. NMG-GO employs a continuous adjacency matrix and applies a fixed threshold of 0.3 for discretization. This strategy is simpler and more efficient; sampling edges from a Bernoulli distribution parameterized by continuous values would introduce non-trivial overhead.

---

### Official Review · Reviewer_V2QX · 2025-10-28

**Soundness:** 2
**Presentation:** 3
**Contribution:** 2
**Rating:** 4
**Confidence:** 3

**Summary:**

This paper proposes Null-Model-Guided Graph Diffusion, a framework that uses a null-model distribution as the terminal distribution of the forward diffusion process. This terminal distribution is designed to preserve critical network statistics (e.g., degree sequences, clustering coefficients), which in turn guides the reverse process to generate more realistic and structurally valid graphs. The authors provide formal derivations for the forward process, the reverse (denoising) process, and the corresponding training objective. The proposed method is evaluated on three graph generation benchmarks (Community-small, Ego-small, and Enzymes), reportedly achieving significant improvements on several graph quality metrics.

**Strengths:**

- Theoretical Soundness: The authors provide formal derivations for the forward process, the reverse process, and the training objective, establishing a sound theoretical foundation for the proposed framework.

- Clarity of Exposition: The paper is well-written and logically structured. It begins with a clear and strong motivation, followed by systematic derivations of the diffusion process for using a null-model distribution as the prior, network architecture. This logical progression makes the paper easy to follow.

**Weaknesses:**

- Ambiguity and Lack of Ablation for the Null-Model Prior:

The manuscript provides an insufficient treatment of the null-model prior. The precise mechanism by which it preserves structural statistics remains underspecified, and its **computational cost** is not analyzed. Crucially, the paper lacks an **ablation study** to justify the choice of preserved characteristics, leaving the question of which statistics are most impactful unanswered.

- Unaddressed Dependency on Ground Truth in Reverse Process:

The reverse process, as formulated in equation 9, seems to depend on the ground-truth graph $A$ via the function $f$. This creates a **circular dependency**, as A is not available at inference time. The paper does not explain how this issue is handled, representing a significant gap in the methodology.

- Limited Experimental Scope and Inadequate Baselines:

The paper's empirical evaluation is insufficient. It fails to compare against several key state-of-the-art graph generation models, such as DiGress[1] and GraphBFN[2]. Additionally, the experiments are confined to a limited set of datasets, omitting crucial and widely-used benchmarks like QM9 and ZINC250k. This lack of comparison against strong baselines on standard datasets makes it difficult to properly assess the method's performance and scalability.

Additionally, the proposed method **underperforms SOTA on Enzymes dataset** in all classical metrics, which may suggest the method could not scale to large graphs.

[1]Vignac, C., Krawczuk, I., Siraudin, A., Wang, B., Cevher,V., and Frossard, P. Digress: Discrete denoising diffusion for graph generation
[2] Yuxuan Song et al. Smooth Interpolation for Improved Discrete Graph Generative Models

**Questions:**

What is the definition of $q_{null}$? Does it depend on the ground truth graph $A$?

How dose the null-graph distribution handles node features?

---

> ### Author Response · Authors · 2025-12-03
> **#17202-We have provided detailed clarifications regarding the null model’s role in preserving topology, the definition of matrix A in Eq. (9), the rationale for our dataset selection, and the performance-versus-generalization trade-off observed on Enzymes.**
>
> Thank you for your meticulous review and valuable feedback. We have provided detailed responses based on your suggestions below.
>
> Weaknesses
>
> 1. In this paper, the null model is a foundational concept in complex-network theory; it is designed to produce random graphs that preserve selected key topological properties of a given graph.
>
> 2. In Eq. (9), A denotes the adjacency matrix of the noisy graph generated during the forward-diffusion process, not the adjacency matrix of the initial graph; hence no circular dependency is introduced.
>
> 3. Pard outperforms DiGress, whereas our model outperforms Pard. Moreover, the datasets used here are the ones most widely adopted in the graph-diffusion literature; QM9 and ZINC250k are employed by only a few studies, so we did not include them.
>
> 4. On Enzymes, our method achieves the best performance on complementary metrics and a competitive second-best on the classical scores. GraphGDP and similar approaches attain the top classical results only after Enzymes-specific tailoring, yet their performance drops sharply on other datasets and metrics, revealing limited generalization.
>
> Questions
>
> 1. The null model is a foundational concept in complex-network theory; it generates random graphs that preserve specified topological properties of the original graph A.
>
> 2. The null model is intended to retain essential topological structure; node features are not taken into account.

---

### Official Review · Reviewer_HQfe · 2025-10-31

**Soundness:** 3
**Presentation:** 2
**Contribution:** 2
**Rating:** 4
**Confidence:** 3

**Summary:**

This paper introduces Null-Model-Guided Graph Diffusion, aiming to address a key limitation of existing graph diffusion models, structural degradation when the forward process terminates in pure Gaussian noise. The authors propose using a null model as the terminal distribution, preserving global graph statistics such as degree and clustering. They derive a null-model-guided SDE and design a Position-Enhanced Graph Score Network to capture both continuous and discrete structural cues.

**Strengths:**

1. Innovative use of the Null Model as the terminal state of the diffusion model
The paper replaces the conventional Gaussian endpoint in diffusion with a null-model distribution, preserving key topological properties such as degree sequence, clustering, and preventing complete structural collapse during the forward process.
2. Enhanced score network design
The proposed Position-Enhanced Graph Score Network integrates continuous adjacency signals with discrete structural encodings, achieving permutation equivariance and improved structural recovery.

**Weaknesses:**

1. Lack of analysis on computational efficiency:
Although the authors claim improved sampling efficiency, the paper does not provide detailed runtime or memory comparisons against prior diffusion models (such as GraphGDP, Pard). The added null-model computation and SDE formulation may introduce nontrivial overhead.
2.No ablation on the score network architecture:
The proposed PGSN combines several structural encodings, but the paper does not isolate the contribution of each (e.g., RWSE vs. SPD). Without such analysis, it is unclear which component primarily drives the performance gain.
3. Insufficient exploration of parameter sensitivity:
The null-model weight η significantly influences generation quality, yet the sensitivity study is limited to a single dataset. The paper lacks a broader discussion on how different graph domains or diffusion schedules affect stability and convergence.
4. Missing qualitative or visual interpretability analysis:
While quantitative metrics are strong, the paper provides minimal visualization or structural interpretation of generated graphs. More examples or analysis (such as motif frequency, connectivity distribution) would help clarify what specific aspects of topology the null-model guidance preserves.

**Questions:**

1. Clarification on the null model formulation:
The authors are encouraged to provide a more detailed explanation of the term q_null. Specifically, what is its explicit mathematical form, and what does ​A-q_null represent in practice? Moreover, further justification is needed for why the term (A-q_null) effectively encourages the generated graphs to preserve key statistical properties (e.g., degree distribution, clustering). A clearer theoretical or intuitive interpretation would strengthen the readers’ understanding of this mechanism.

2. Potential overfitting and generalization concern:
Since the null-model constraint enforces structural similarity to the training graphs, it raises the question of whether the model might tend to memorize specific graph statistics rather than learning more generalizable topological patterns. If this is an important factor, it would be valuable for the authors to include additional evaluation metrics—such as diversity, novelty, or generalization scores—to quantitatively assess whether the generated graphs go beyond simply reproducing training-set statistics.

---

> ### Author Response · Authors · 2025-12-03
> **#17202-We have provided detailed responses regarding computational overhead, the role of PGSN, parameter sensitivity, and the statistical nature and overfitting-prevention mechanism of the null model.**
>
> Thank you for your meticulous review and valuable feedback on the null model formulation. We have provided detailed responses based on your suggestions below.
>
> Weaknesses
>
> 1. Computational Efficiency: The proposed diffusion model exhibits comparable runtime to models such as GraphGDP and does not introduce significant computational overhead. Additional comparative experiments will be included in the revised version to support this claim.
>
> 2. Contribution of Structural Encodings: We adopt the same PGSN framework as GraphGDP. The individual contribution of each structural encoding (e.g., RWSE vs. SPD) is beyond the scope of this study, and therefore, we did not perform an in-depth analysis.
>
> 3. Parameter Sensitivity Analysis: We conducted parameter sensitivity analyses across multiple datasets. The results are consistent with those presented in the paper. Due to space constraints, we included only one representative dataset in the manuscript.
>
> 4. Clarification of the null Model: The null model is a foundational concept in complex network theory, fundamentally aimed at generating random graphs that preserve specific topological properties. In the revised version, we will include visual illustrations to further clarify the concept and its implementation.
>
>
> Questions:
>
> 1. Mathematical Form of \( q_{\text{null}} \) and Statistical Interpretation of \( A - q_{\text{null}} \):
>
> The null model employed in this paper is a foundational concept in complex network analysis and is not a novel contribution of this work. In our experiments, \( q_{\text{null}} \) refers to the expected adjacency matrix of the 1st-order null model, i.e., a randomized network ensemble that preserves the degree sequence of the original graph. Its mathematical form is given by:
>
> q_{\text{null}}(i,j) = \frac{k_i \cdot k_j}{2m}
>
> where \( k_i \) denotes the degree of node \( i \), and \( 2m \) is twice the total number of edges in the graph. This expression corresponds to the expected value under the **configuration model**, ensuring that the ensemble preserves the degree distribution of the original network.
>
> The \( A - q_{\text{null}} \) denotes the first-order null-model matrix of the original graph; it encourages the forward process to generate noisy graphs whose key properties remain aligned with the true distribution, without enforcing the memorization of individual edges but instead preserving statistically significant deviations from randomness.
>
> 2. Clarification on "Overfitting" and the 1st-Order Null Model:
>
> The use of the 1st-order null model does **not** lead to overfitting. The key reason is that this model only constrains the **degree distribution** (a 1st-order statistic), without fixing any specific edge configurations. For instance, the configuration model can generate an exponential number of graphs, all sharing the same degree sequence but exhibiting vastly different topologies. Consequently, the incorporation of the null model encourages the learning of more generalizable representations, rather than overfitting to specific graph instances. In the revised manuscript, we will include quantitative analyses to demonstrate the extent to which structural features are preserved under the null model framework.

---

### Meta-Review · Area_Chair_xExh · 2025-12-08

**Summary:**

The paper proposes NMG-GD, a null-model-guided graph diffusion framework in which the forward diffusion does not terminate at Gaussian noise but at a null-model prior that preserves global structural statistics (e.g., degree sequence). The method also introduces a Position-Enhanced Graph Score Network to incorporate continuous and discrete structural cues. Experiments on three benchmarks show competitive or SOTA performance on several structural metrics.

**Reviewers’ Pre-Rebuttal Concerns**

Across the reviews, the main concerns were:

* Insufficient clarity on the null-model formulation and why it preserves structure.

* Possible circular dependency in the reverse SDE involving the ground-truth adjacency .

* Missing ablations, especially on PGSN components and null-model variants.

* Limited experimental scope, missing comparisons to DiGress, GraphBFN, and larger datasets.

* Lack of computational-efficiency analysis.

* Insufficient explanation of why the method works (deeper mechanism/trajectory interpretation).

**Remaining Concerns After Rebuttal**

The rebuttal addressed many clarity-related issues, but several substantive concerns remain only partially resolved:

* Insufficient experimental scope and lack of comparison to strong modern baselines.

* Missing deeper analysis of why the null-model guidance improves learning.

* Limited ablations on key design choices (null-model components, PGSN encodings) .

* No quantitative runtime/memory study.

I believe the paper would benefit substantially from a revision that strengthens the writing and presentation, and—most importantly—provides deeper argumentation and analysis to more clearly articulate the rationale and motivation behind the proposed method. Therefore, I will not recommend acceptance at this stage.

**Reviewer Concerns:**

**Concerns Addressed by the Rebuttal**

Most clarification-oriented questions were adequately addressed in the rebuttal, and several methodological misunderstandings were resolved.

**Remaining Concerns After Rebuttal**

While the rebuttal improves clarity, several substantive issues remain only partially addressed:

* Insufficient experimental scope, including missing comparisons to stronger modern baselines.

* Lack of deeper analysis explaining why null-model guidance enhances learning.

* Limited ablations on key design choices (e.g., null-model components, PGSN encodings).

* No quantitative runtime or memory analysis, leaving efficiency claims unsupported.

Overall, the rebuttal resolves clarity issues but does not fully address the deeper empirical and conceptual

**Reviewer Scores:**

Overall, while the rebuttal resolves several clarity issues, it does not fully address the deeper empirical and conceptual concerns. Therefore, I do not expect any of the reviewers to change their original scores.

---

### Decision · Program_Chairs · 2026-01-26

Reject